# Effect of Solidification Time on Microstructural, Mechanical and Fatigue Properties of Solution Strengthened Ferritic Ductile Iron

**Thomas Borsato [1], Paolo Ferro [1,*] , Filippo Berto [2] and Carlo Carollo [3]**

1   Department of Engineering and Management, University of Padova, Stradella S. Nicola, 3,
    I-36100 Vicenza, Italy; thomas.borsato@phd.unipd.it
2   Department of Engineering Design and Materials, NTNU, Richard Birkelands vei 2b,
    7491 Trondheim, Norway; filippo.berto@ntnu.no
3   VDP Foundry SpA, via Lago di Alleghe 39, 36015 Schio, Italy; carollo@vdp.it
*   Correspondence: paolo.ferro@unipd.it; Tel.: +39-0444-998769

**Abstract:** Microstructural, mechanical, and fatigue properties of solution strengthened ferritic ductile iron have been evaluated as functions of different solidification times. Three types of cast samples with increasing thickness have been produced in a green sand automatic molding line. Microstructural analyses have been performed in order to evaluate the graphite nodules parameter and matrix structure. Tensile and fatigue tests have been carried out using specimens taken from specific zones, with increasing solidification time, inside each cast sample. Finally, the fatigue fracture surfaces have been observed using a scanning electron microscope (SEM). The results showed that solidification time has a significant effect on the microstructure and mechanical properties of solution strengthened ferritic ductile iron. In particular, it has been found that with increasing solidification times, the microstructure becomes coarser and the presence of defects increases. Moreover, the lower the cooling rate, the lower the tensile and fatigue properties measured. Since in an overall casting geometry, same thicknesses may be characterized by different microstructures and mechanical properties induced by different solidification times, it is thought that the proposed methodology will be useful in the future to estimate the fatigue strength of cast iron castings through the numerical calculation of the solidification time.

**Keywords:** silicon solution strengthened ferritic ductile iron; thickness; solidification time; microstructure; mechanical properties; fatigue; thermal analysis

## 1. Introduction

Since 2012, solution strengthened ferritic ductile irons (SSF-DI) have been introduced in the UNI EN 1563 standard [1]. These alloys are characterized by a fully ferritic matrix, reinforced by the addition of a balanced amount of Silicon, which provides a combination of high strength and ductility [2].

Two of the most important microstructural parameters that are widely used for the estimation of the quality of the ductile iron castings are nodule count and nodularity, as described in the ASTM E2567-16a standard [3]. By increasing the solidification time, the number of graphite particles with a spheroidal shape decreases and their dimension increases; consequently, the mechanical properties are affected. In particular, it has been found that by increasing the section thickness, the nodule count decreases, while the ferrite content increases. Under these conditions, tensile strength and hardness decrease, while elongation at failure increases [4–6].

Longer solidification times cause an increased risk of formation of microstructural defects, with detrimental effect on the mechanical properties [7]. The most common defects that can be found in castings are non-metallic inclusions, shrinkage porosities, and undesired segregation or graphite particles that deviate from the spheroidal shape. Among degenerated graphite morphologies, the branched and interconnected particles, known as chunky graphite (CHG), spiky, exploded, and compacted graphite are the most important and detrimental considering their influence on the mechanical properties [8].

All these defects can be only partly avoided through the optimization of the production process. For example, it was found that adjusted post-inoculation could decrease the dimensions of microshrinkage porosities in heavy section castings [9,10].

Several works [11,12] studied the factors affecting the graphite degeneration in thick walled castings, with particular attention to chunky graphite, which is the most frequent. Although CHG has been considered in many studies, no generally accepted theory for its formation has been found yet. What is known is its detrimental effect on the mechanical properties, with particular reference to the ultimate tensile strength and ductility; but CHG seems also to affect the crack propagation stage during fatigue loadings [13–15]. While in the case of spheroidal graphite particles, decohesion between the nodules and the metal matrix happens, the crack propagates easily through the chunky graphite areas.

On the other side, it has been found by many researchers that the crack initiation stage is more affected by microstructural defects such as non-metallic inclusions, microshrinkage porosities or spiky graphite [16–23]. For example, Borsato et al. [18] proposed a new equation for the assessment of the fatigue limit of ductile iron castings basing on the defects dimensions and the static mechanical properties of the analyzed alloy. In their experiments about the fatigue strength of heavy section ductile iron castings, both Foglio et al. [16] and Ferro et al. [19] observed that the crack initiating defect was a porosity in the vicinity of the sample surface.

While a great effort was spent in the past in order to characterize the traditional ductile irons with ferritic and/or pearlitic matrices, limited data is available in technical papers regarding the new generation ductile irons (SSF-DI) [14,17,24]. Furthermore, only the mechanical properties versus thickness correlation is found, the main drawback of which is that equal thicknesses do not mean necessary equal microstructure and thus mechanical properties. This is the reason why an attempt has been made in the present work to correlate the static and fatigue properties of a solution strengthened ferritic ductile iron with the solidification time supposed to be the most important microstructure-influencing parameter.

## 2. Experimental Procedure

In this paper, a solid solution strengthened ferritic ductile iron with 3.25 wt % Si was investigated. Melt was prepared in medium frequency induction furnace from pig iron, steel scrap, and ductile cast iron returns. The spheroidizing treatments and the preconditioning were performed in a dedicated ladle by adding 1.2% FeSiMg commercial alloy, containing 1 wt % RE, and 0.3% inoculant (75 wt % Si, 1 wt % Ca) using the sandwich method. After the spheroidizing and inoculation process, and just before pouring the iron into the molds, a metal sample was analyzed by optical emission spectrometry to determine the chemical composition. At the same time, in order to complete the alloy characterization, a thermal analysis was carried out. A standard cup for the thermal analysis, containing the same weight percentage of inoculant of the castings was filled and the cooling curve was then measured by using ITACA MeltDeck$^{TM}$ (ProService Tech, Borgoricco (PD), Italy).

The final chemical composition of the material is shown in Table 1.

**Table 1.** Final chemical composition (wt %).

| C | Si | S | P | Mn | Mg | Ce | Ceq |
|------|------|-------|-------|------|-------|--------|------|
| 3.31 | 3.25 | 0.008 | 0.025 | 0.13 | 0.050 | 0.0018 | 4.40 |

The carbon content was chosen to be 3.3 wt %, in order to maintain a near eutectic composition, with the Equivalent Carbon calculated using the equation: $C_{eq} = C\% + 0.33(Si\% + P\%)$ [25].

With the aim to evaluate the effect of increasing solidification times on microstructural and mechanical properties, three different geometries, with increasing section thickness were produced in a green sand automatic molding line. 15 molds were produced, each of them containing cast samples with geometries taken from the UNI EN 1563:2012 standard [1]. In particular, the round bar-shaped (type b), the Y-shaped type III and the Y-shaped type IV were used, the relevant wall thicknesses of which were 25, 50, and 75 mm respectively.

From the round bar samples, only tensile test specimens were machined, while from the Y-shaped samples fatigue specimens were also obtained. In particular, the tensile test specimens were characterized by a net diameter of 14 mm, while the fatigue specimens had a rectangular net cross section of $10 \times 15$ mm$^2$.

In order to evaluate the influence of increasing solidification time on the mechanical properties, specimens were taken from the three different cast samples. In particular, in the case of round bar shaped samples with a diameter of 25 mm, the tensile specimens were directly machined. Differently, in the case of Y-shaped samples, a block of about $25 \times 25 \times 175$ mm$^3$ was cut before the final machining. The positions where the specimens were taken from are shown in Figure 1; it can be noted that from each type IV sample, two specimens were obtained.

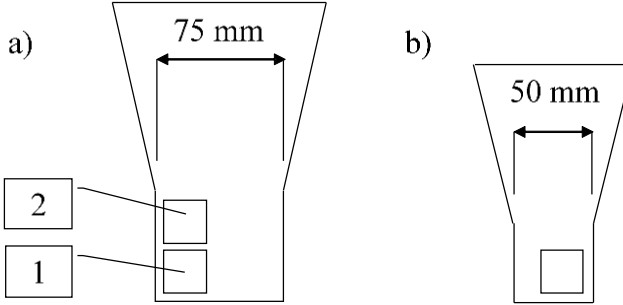

**Figure 1.** Position of specimens for tensile tests (according to UNI EN 1563:2012) taken from the Y-shaped type IV (**a**) and type III (**b**).

In order to compute the solidification time within each cast sample, numerical analyses were carried out by using the code Novaflow & Solid®(Version 4, Novacast, Ronneby, Sweden). The temperature dependent material properties for ductile iron and green sand have been taken from the database of the numerical code. A size element of 2.5 mm was used for the mesh. The temperature history measured by a virtual thermocouple positioned at the centre of each zone (Figure 1) was used to calculate the solidification time of the whole zone. Tensile tests have been conducted at room temperature according to ISO 6892-1:2016 [26] by using the INSTRON 5500R (Instron, Norwood, MA, USA) tensile test machine under strain rate control. Fatigue tests have been performed according to ASTM E468-18 Standard. A resonant testing machine (RUMUL Testronic 150kN, Russenberger Prüfmaschinen AG, Neuhausen am Rheinfall, Schweiz) was used with a sinusoidal tensile pulsating load at the frequency of about 130 Hz and nominal load ratio R = 0. Tests have been stopped at the total separation of the two parts of the specimens, or after reaching $10^7$ cycles. The staircase method was carried out with an applied stress increment of 10 MPa in order to evaluate the fatigue strength corresponding to a fatigue life of $10^7$ cycles.

Fatigue tests have been conducted on plain specimens taken from Y-shaped type III and IV cast samples. In particular for each sample, specimens were taken from three levels, numbered consecutively from 1 to 3 going towards the thermal center of the casting, as shown in Figure 2.

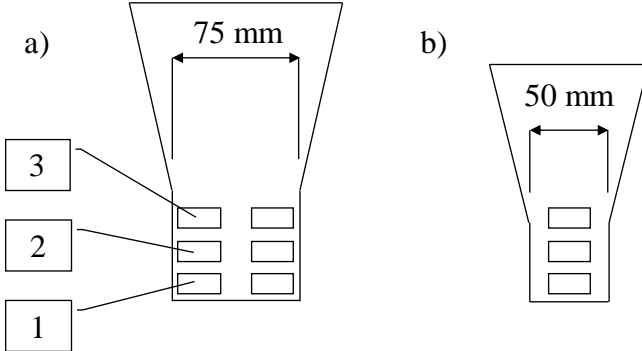

**Figure 2.** Position of fatigue specimens (rectangular cross section) taken from the Y-shaped type IV (**a**) and type III (**b**).

In the case of type IV sample, six specimens have been obtained, while, due to the smaller dimension, it was possible to take only three specimens from each type III sample. A total of 60 and 30 specimens have been tested for type IV and type III sample, respectively.

Microstructural parameters, such as nodule count, nodule size, nodularity and matrix structure have been evaluated according to the ASTM E2567-16a standard, using an optical microscope and an image analysis software, on polished samples in the unetched and etched (Nital 5%) conditions.

Finally, the fracture surfaces of some broken specimens under fatigue loading have been examined using a scanning electron microscope (Quanta 2580 FEG, FEI, Boston, MA, USA).

## 3. Results

### 3.1. Thermal Analysis

The result of thermal analysis is the cooling curve of the alloy from which it is possible to calculate the liquidus temperature, the minimum and maximum eutectic temperature, the temperature at the end of solidification, and finally the temperatures during the solid phase transformation [27,28].

The cooling curve of the SSF-DI obtained from the standard cup and its first time derivative are shown in Figure 3. From a global point of view, two plateaus during the eutectic and the solid-state transformation can be observed.

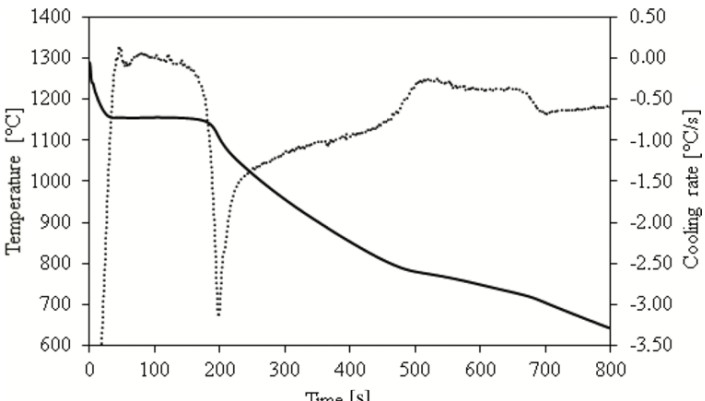

**Figure 3.** Cooling curve (solid line) and its first derivative (dotted line) of the solution strengthened ferritic ductile iron studied in this work.

From the comparison between the curves of traditional (pearlitic and ferritic-pearlitic) cast irons and solution strengthened ferritic ductile irons during the eutectic and the eutectoid transformation (Figures 4 and 5, respectively), it is visible that there are little variations in the solidification

behavior (minimum eutectic temperature about 1153 °C), while the main difference is related to the solid-state transformation.

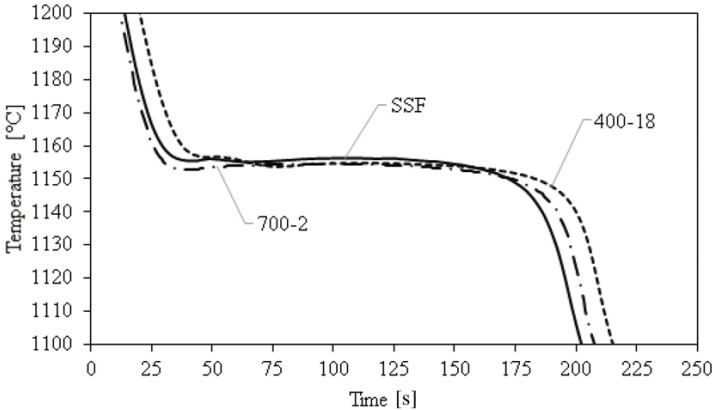

**Figure 4.** Comparison of temperature profile during the eutectic transformation between traditional and solution strengthened ferritic ductile iron.

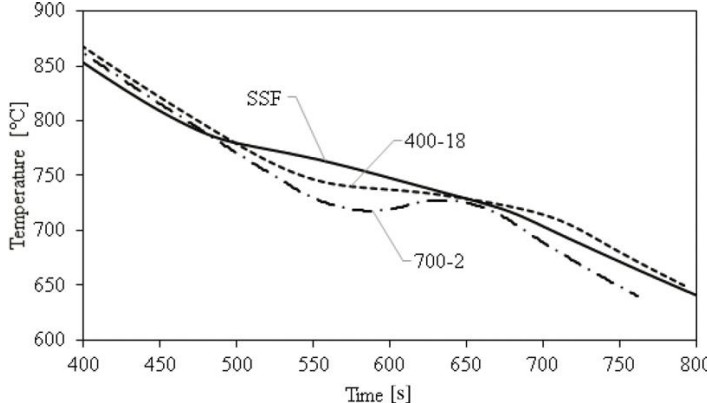

**Figure 5.** Comparison of temperature profile during the solid-state transformation between traditional and solution strengthened ferritic ductile iron.

As reported in literature [29] the different chemical compositions of the alloys promote the formation of pearlitic and/or ferritic matrix, with differences on the eutectoid temperature and on the cooling rate. In particular, the cooling curve of the GJS 700-2 showed an arrest and a recalescence that is associated to the formation of pro-eutectoid cementite present in pearlite, while in the case of GJS 400-18, the ferritic matrix leads to a higher eutectoid temperature.

Finally, due to the higher silicon content, the eutectoid temperature of the solution strengthened ductile iron is increased with respect to the traditional ferritic grade [30].

*3.2. Tensile Tests*

Five specimens for each of the four conditions have been tested. Table 2 summarizes the mean values of the results obtained as a function of section thickness and corresponding solidification time, obtained from the numerical analysis.

Compared to other methods used in literature [31], it is important to highlight the fact that, through the solid solution strengthening made by Silicon, it is possible to reach both high strength and ductility. In particular, the ultimate tensile strength is similar to the value of traditional GJS 500-7, while the elongation at failure is close to that of GJS 400-18 cast iron [1].

As expected, by increasing the solidification time, the mechanical properties decrease. This is visible not only between specimens taken from the three different kinds of cast samples, but also

from different positions within the same sample. As a matter of fact, the longer the solidification time, the higher the nodules count and nodularity (as described in the paragraph 3.4) and the higher the mechanical properties of the alloy. It means that the solidification time, can be considered as a promising parameter useful to estimate the mechanical properties of the alloy, provided that the constitutive relation 'solidification time versus mechanical property' is experimentally determined.

**Table 2.** Tensile test results (mean values and standard deviation (in brackets)) of specimens taken from cast samples as a function of section thickness and corresponding solidification time.

| Cast Sample | Thickness [mm] | Solidification Time [min] | $\sigma_{UTS}$ [MPa] | $\sigma_{y\ 0.2\%}$ [MPa] | $\varepsilon_R$ % |
|---|---|---|---|---|---|
| Round bar shaped type b | 25 | 2.5 | 507 (0.5) | 395 (0.5) | 19.8 (0.1) |
| Y-shaped type III | 50 | 9.8 | 492 (1.0) | 389 (0.6) | 17.1 (0.1) |
| Y-shaped type IV (1) | 75 | 16.2 | 487 (0.6) | 386 (0.6) | 17.2 (0.9) |
| Y-shaped type IV (2) | 75 | 22.1 | 468 (2.1) | 375 (1.2) | 13.0 (1.0) |

### 3.3. Fatigue Tests

The results of the fatigue tests have been statistically analyzed by using a method that estimates the full S-N curve by considering both the finite life and the run out specimens according to ISO 12107:2012 standard [32]. It is assumed that the finite fatigue life regime consists of an inclined straight line in a logarithm scale with data following a log-normal distribution, while the fatigue endurance region is represented by a horizontal line.

In order to have a graphical comparison of the fatigue behavior, the specimens taken from the different zones within the cast samples are represented with different symbols in Figure 6. The lines represent the 50% fatigue survival probability estimations.

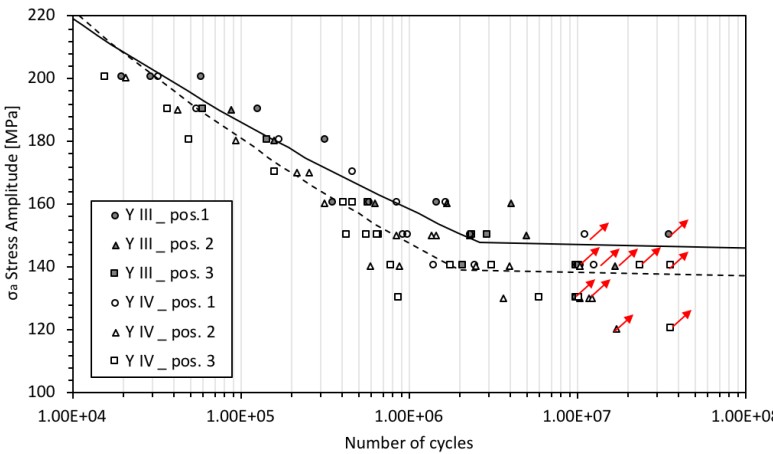

**Figure 6.** Fatigue life of specimens taken from different position within the cast samples. Solid line and dotted line represent the estimated fatigue curves at 50% survival probability for Y-shaped type III and type IV cast samples respectively. Run out specimens marked with an arrow.

Firstly, the statistical analysis of fatigue data has been performed considering all together the specimens taken from the same cast sample geometry.

In particular it can be observed in Table 3 that, by increasing the section thickness of the casting (going from 50 to 75 mm) the fatigue stress amplitude ($\sigma_a$) corresponding to 50% survival probability is lowered by about 7 MPa while the scatter index $T_\sigma$, defined as the ratio between the stress amplitude at 10% and 90% survival probability, increases.

Moreover, it can be observed from the graph in Figure 6 that the specimens taken from the position number 1 of the type IV cast samples behave in a similar manner to those obtained from the Y III samples.

**Table 3.** Fatigue strength at 50% survival probability and scatter index at $10^7$ cycles considering cast samples thickness.

| Cast Sample | Thickness [mm] | $\sigma_a$ [MPa] | $T_\sigma$ |
|---|---|---|---|
| Y-shaped type III | 50 | 145 | 1.13 |
| Y-shaped type IV | 75 | 138 | 1.30 |

In order to better understand the fatigue behavior of the castings, data have been re-analyzed considering no more the thickness, but the solidification time. In particular, three time ranges have been defined based on the numerical simulation results shown in Table 4. Under these conditions, samples taken from position number 3 of the type III sample and from position 1 of the type IV sample, which fell within the same solidification time range, have been analyzed together.

**Table 4.** Solidification times in the positions shown in Figure 5 obtained from the numerical analysis. The value read at the node in the center of each sample was considered.

| Position | Y-Shaped Type III | Y-Shaped Type IV |
|---|---|---|
| 1 | 8.5 min | 14.5 min |
| 2 | 11.3 min | 18.1 min |
| 3 | 14.5 min | 22.1 min |

The obtained fatigue curves at 50% survival probability (the probability that the sample will break at lower load amplitude values) are shown in Figure 7, while the estimated fatigue strength and the scatter index are reported in Table 5 for each range.

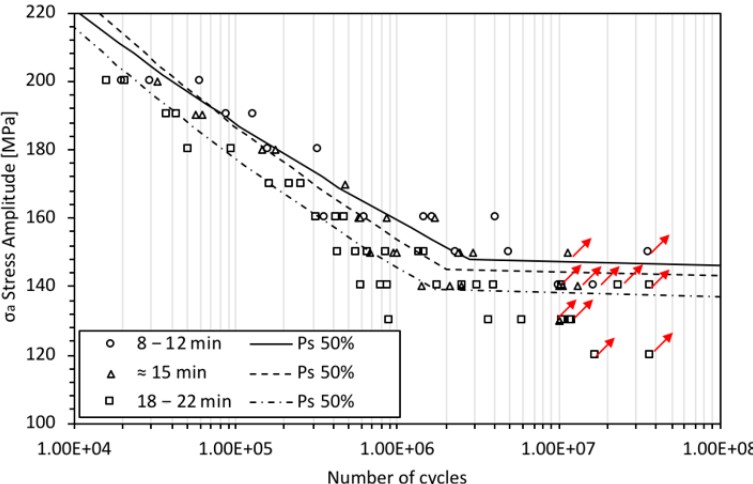

**Figure 7.** Fatigue life of specimens as a function of solidification time. Lines represent the estimated 50% survival probability curve in the defined solidification time ranges. Run out specimens marked with an arrow.

**Table 5.** Fatigue strength at 50% survival probability and scatter index at $10^7$ cycles considering solidification time range.

| Solidification Time | $\sigma_a$ [MPa] | $T_\sigma$ |
|---|---|---|
| 8–12 min | 147 | 1.10 |
| ≈15 min | 143 | 1.23 |
| 18–22 min | 136 | 1.25 |

It is important to note that, taking into account the solidification times, it is possible to obtain a more accurate fatigue life estimation in the different positions within the castings, with reduced

scatter index compared to the results achieved by considering only the section thickness. This is due to the fact that the statistical analyses were carried out by taking into account specimens with similar cooling conditions, which lead to reduced variability in the mechanical properties.

The relationship between positions within the cast sample and the calculated solidification time is shown in Table 3.

### 3.4. Microstructure and Fractography

A summary of the microstructural properties is presented in Table 6. In Figure 8, the microstructure of the round bar shaped cast sample is shown, where it can be noted as a fully ferritic matrix with high number of nodules. It can be observed that the solidification time influences the microstructure of the alloys; in particular, it was confirmed that decreasing the cooling rate, fewer nodules are formed with gradually increasing dimensions and lower nodularity. Specimens taken from different cast samples but with the same cooling conditions are characterized by similar microstructural parameters, such as nodule count and nodule diameter.

**Table 6.** Microstructural properties of samples. Mean values and standard deviation (in brackets).

| Cast Sample | Position | Nodule Count [Nodules/mm$^2$] | Mean Nodule Diameter [μm] | Nodularity |
|---|---|---|---|---|
| Round bar | - | 304 | 19 (5.5) | 93% |
| Y III | 1 | 115 | 29 (9) | 85% |
| | 2 | 98 | 30 (11) | 80% |
| | 3 | 90 | 28 (10) | 77% |
| Y IV | 1 | 95 | 29 (9.5) | 82% |
| | 2 | 84 | 31 (11) | 76% |
| | 3 | 63 | 33 (13) | 70% |

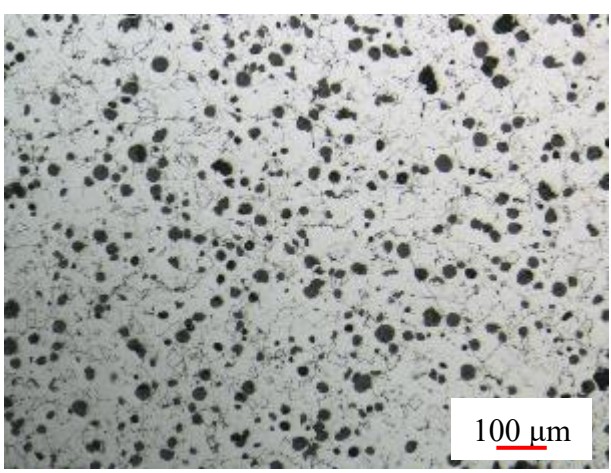

**Figure 8.** Micrograph of specimen taken from round bar shaped cast sample, etched with Nital 5%.

The increasing in the solidification time could promote also the formation of microstructural defects such as microshrinkage porosities, segregation, or degenerated graphite particles. In fact, going toward the center of the cast samples, it has been found an increasing amount of areas containing branched and interconnected graphite particles, classified as chunky graphite (Figure 9).

Moreover, in the longer to solidify zones, microshrinkage cavities and small area of pearlite, due to segregation of carbide promoter or pearlitizing elements, have been found.

The microstructural properties confirmed what observed during the mechanical tests.

Increasing the section thickness, the solidification time increases and the undercooling decreases. Under these conditions, the number of graphite nodules is lowered, while the risk of forming microstructural defects (microshrinkage porosities, segregations, graphite degenerations, etc.) increases.

The lower nodule count and the presence of such defects could be related to the lower mechanical properties and the higher fatigue scatter index of specimens taken from longer to solidify zones.

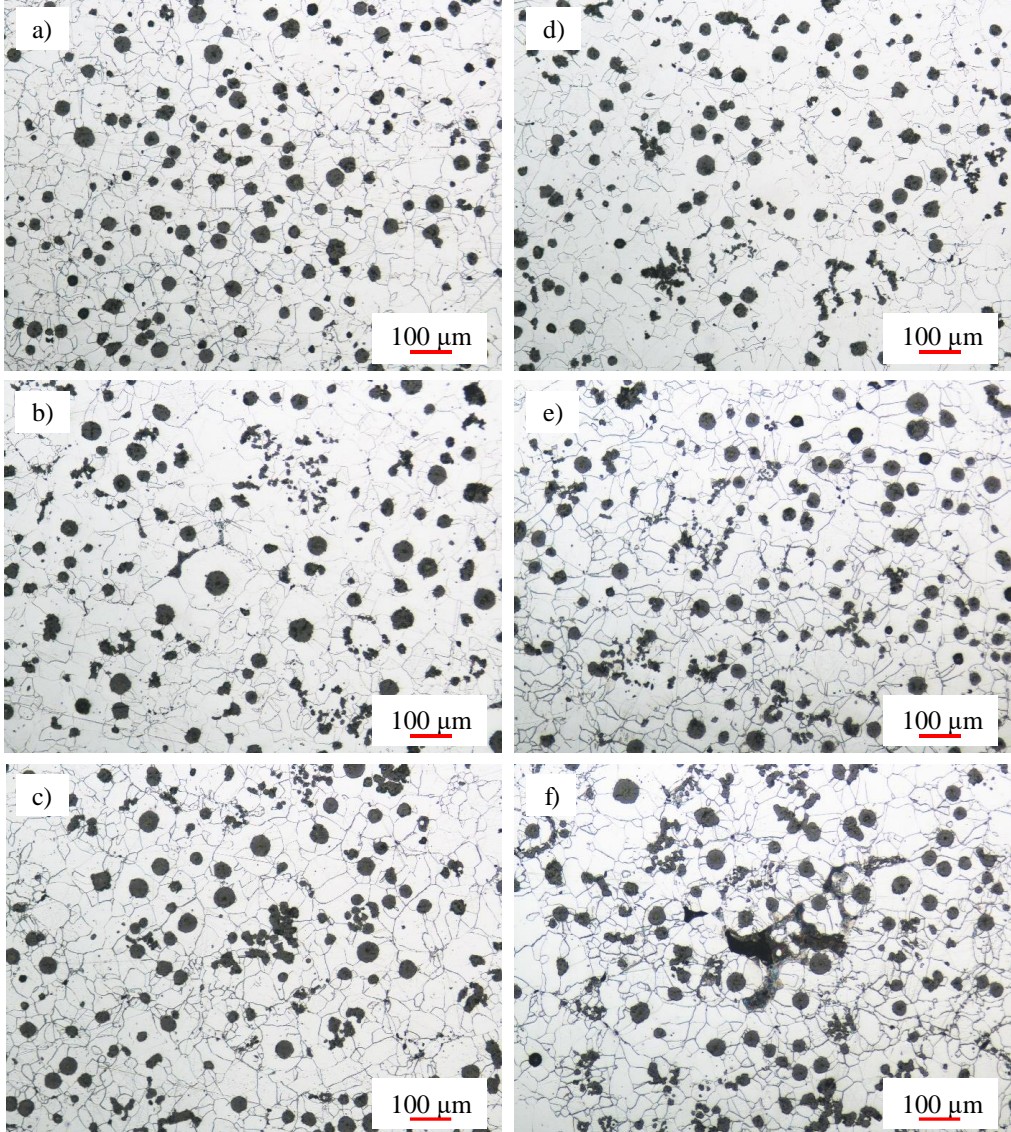

**Figure 9.** Micrographs of specimens taken from type III cast sample, position 1 (**a**), 2 (**b**) and 3 (**c**) and type IV cast sample, position 1 (**d**), 2 (**e**) and 3 (**f**).

The analyses of fracture surfaces of some fatigue broken specimens revealed that the failure initiated at microshrinkage porosities located near the free surface of the specimens (Figure 10a,b).

The dimensions of such initiating defects ($\sqrt{area}$ parameter) increase with the solidification time, passing from less than 100 μm to more than 400 μm.

It was also found that the failure seems to happen in different ways. Most of the fracture surfaces showed a ductile dimple fracture with microvoids coalescence (Figure 10c,d), however, some areas with brittle transgranular cleavage and intergranular fracture have been even detected (Figure 10e).

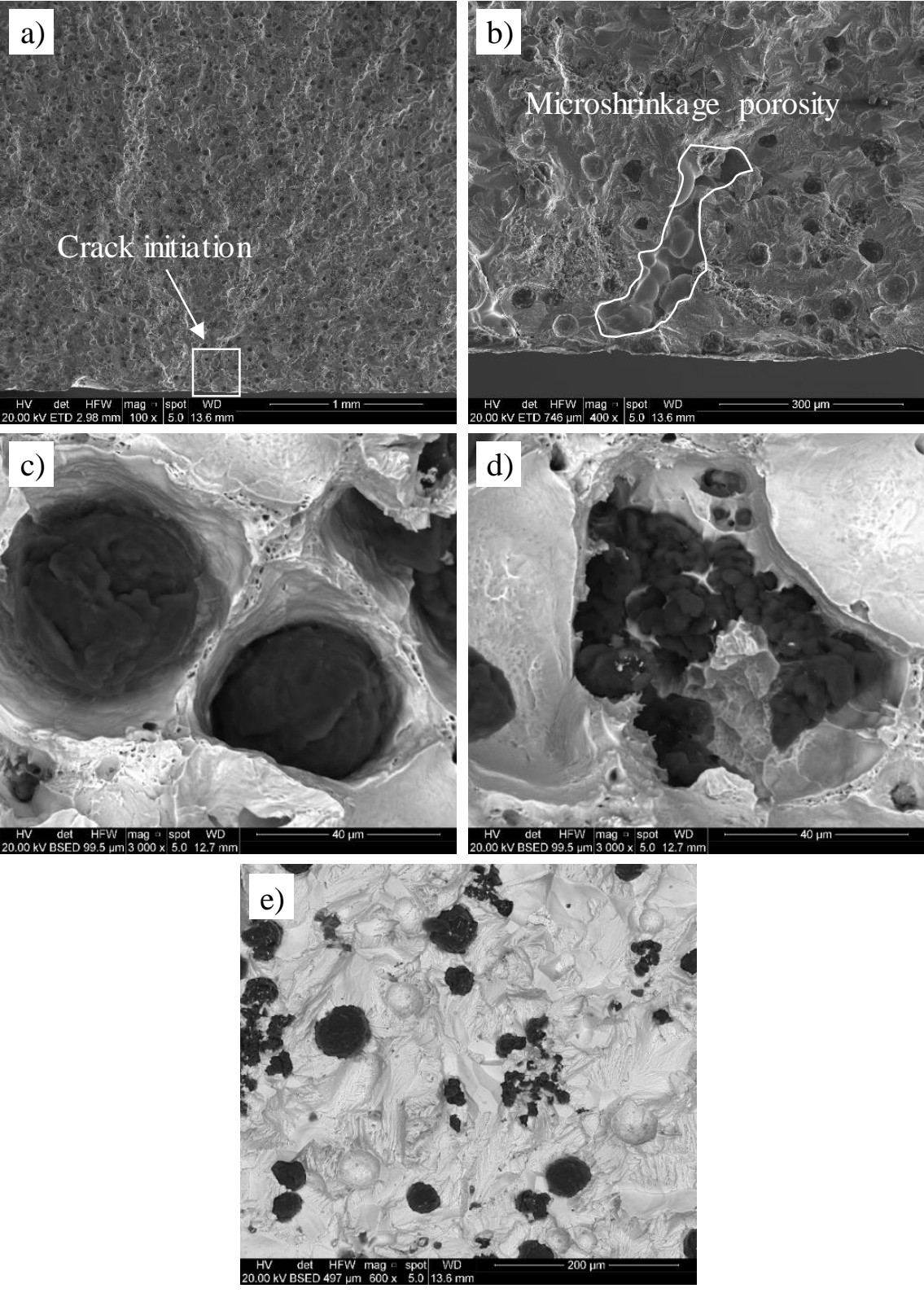

**Figure 10.** Scanning electron microscope (SEM) images of fracture surfaces showing a panoramic overview of the crack initiation and propagation zone (**a**) and a particular of the crack initiating defect (microshrinkage porosity) (**b**); dimple fracture with microvoids coalescence in the presence of spheroidal graphite nodules (**c**) and degenerated graphite particles (**d**); image of the coexistence of brittle transgranular cleavage and intergranular fracture (**e**).

## 4. Conclusions

In this paper, the microstructural, mechanical and fatigue properties of a solution strengthened ferritic ductile cast iron characterized by different solidification times have been investigated. It has been confirmed that low cooling rates affect the strength and the ductility of the castings. Due to the high amount of silicon, the microstructure exhibits a fully ferritic matrix, with only small areas of pearlite, due to the segregation of undesired elements, in the thicker sections. Moving toward the zone that takes longer to solidify, the number of graphite nodules decreases and an increasing amount of degenerated chunky graphite is found. The fatigue behavior has been evaluated using specimens taken from different zones inside the castings within defined solidification time ranges. It was confirmed that a better estimation of the fatigue life is achievable by considering the solidification times rather than the section thickness. A decrease of the fatigue strength and an increase of the scatter index were observed with increasing the solidification times, due to the increase in the defects dimensions.

All the samples showed shrinkage porosities, which have been identified as crack initiation sites. The fracture surface revealed that the crack propagates easier through the areas with chunky graphite compared to areas with spheroidal graphite nodules. Most of the fracture surface showed a ductile dimple fracture with microvoids coalescence, but some areas with brittle transgranular cleavage and intergranular fracture were also detected. Finally, it was observed that the solidification time can be really considered as a microstructure-influencing parameter. Basing on the intrinsic relation between microstructure and mechanical properties, it will be possible in the future to estimate the static and fatigue strength of the alloy simply by a numerical computation of the solidification time, provided that the constitutive relation 'solidification time versus mechanical property' is 'a-priori' determined.

**Author Contributions:** Conceptualization, P.F., T.B. and F.B.; methodology, P.F. and T.B.; validation, T.B.; formal analysis, T.B.; investigation, T.B. and C.C.; F.B.; writing—original draft preparation, T.B.; writing—review and editing, P.F. and F.B.; visualization, P.F., T.B. and F.B.; supervision, P.F. and F.B.

**Funding:** This research received no external funding.

**Acknowledgments:** The authors would like to thank VDP Fonderia SpA for the material supply and technical support of the project.

**Conflicts of Interest:** The authors declare no conflict of interest.

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
