# Peer review of "Effect of Solidification Time on Microstructural, Mechanical and Fatigue Properties of Solution Strengthened Ferritic Ductile Iron"

_metals, doi:10.3390/met9010024_

Reviewer 1 Report

The article represents an experimental characterization of a particular family of spheroidal cast irons, the SSFs that are investigated in a good detail. Tensile tests and fatigue tests are proposed, together with thermal and metallographic analyzes. Also numerical estimates on solidification times are provided. 

The complexity of the experiment (e.g. number of samples analyzed) is not particularly high, but adequate to meet the standards. The topics are presented with care and substantial precision with a suggestion of minor changes as indicated in the attached document.

 At the same time, some central aspects should be considered with the scope to improve the overall merit of the paper:

- introduction is quite generic and, especially in the initial parts, quite redundant considering the scientific focus of the Journal. 

- experimental results are often presented, without analyzing the problem of the variability of data, usually quite strong in the determination of cast iron properties.

- it is sometimes unclear if there is any underlying reason for specific choices and assumptions. For instance, for which reason these specific alloy and process parameters have been preferred to others.

- it is almost unclear what is the novelty of the research and results. For which reason this article is special or different respect to others available in the scientific literature  Also the conclusions express the fact the results, more or less, confirm everything already known. Thus, it is suggested to extend introduction and conclusion reinforcing the description of the novelty.

- one third of references are related of researches performed by the Authors. Sometimes, these references are also used to confirm methods and results.It would be useful to collect & describe Authors' research action and main results with the scope to demonstrate how this specific research is part, integrated or different respect to the others.

- Considering the relevance of the topic, leading to an large literature, an extension of the bibliography is suggested. Contributions from METALS are particularly welcomed, with the scope to show how this topic is considered as valid and has been approached inside the journal.

Author Response

We really thank the Reviewer for the time spent in reading our work and comments aimed to improve it. All the suggestions were taken into account and the paper modified accordingly 

Reviewer #1: 
The article represents an experimental characterization of a particular family of spheroidal cast irons, the SSFs that are investigated in a good detail. Tensile tests and fatigue tests are proposed, together with thermal and metallographic analyses. Also numerical estimates on solidification times are provided. 

The complexity of the experiment (e.g. number of samples analysed) is not particularly high, but adequate to meet the standards. The topics are presented with care and substantial precision with a suggestion of minor changes as indicated in the attached document.

Reply: Authors thank a  lot the Reviewer for his very good revision and  did their best for improving the manuscript according to the Reviewer’s suggestions attached to the document provided by him.

 At the same time, some central aspects should be considered with the scope to improve the overall merit of the paper:

- introduction is quite generic and, especially in the initial parts, quite redundant considering the scientific focus of the Journal. 

Reply: we agree with the reviewer’s observation, thus the first part of the introduction was simplified. The introduction was even more focused on the novelty of the work

- experimental results are often presented, without analysing the problem of the variability of data, usually quite strong in the determination of cast iron properties.

Reply: the Standard deviations are specified in metallurgical data and fatigue results were statistically elaborated. In the revised version of the paper the standard deviations were also added to mechanical results

- it is sometimes unclear if there is any underlying reason for specific choices and assumptions. For instance, for which reason these specific alloy and process parameters have been preferred to others.

Reply: as specified in the introduction, we have chosen SSF-DI because of its increasing demand  in the market and limited data available in technical papers. Process parameters are those required to produce heavy-sections castings as required by the market. 

- it is almost unclear what is the novelty of the research and results. For which reason this article is special or different respect to others available in the scientific literature  Also the conclusions express the fact the results, more or less, confirm everything already known. Thus, it is suggested to extend introduction and conclusion reinforcing the description of the novelty.

Reply: we thank the Reviewer for that observation. The novelty  of the paper consists in heaving determined for the first time  and for this kind of alloy the correlation between the mechanical properties and the solidification times. This will overcome the limitations implicit in the mechanical properties-thickness relationship as specified in the updated version of the abstract, introduction and conclusions.

- one third of references are related of researches performed by the Authors. Sometimes, these references are also used to confirm methods and results. It would be useful to collect & describe Authors' research action and main results with the scope to demonstrate how this specific research is part, integrated or different respect to the others.

Reply: As described in the introduction, the present work is a  natural extension of the previous ones in which new static and fatigue data about SSF-DI are measured and correlated for the first time to the solidification time. Some explicit references descriptions were added to the text as suggested.

- Considering the relevance of the topic, leading to an large literature, an extension of the bibliography is suggested. Contributions from METALS are particularly welcomed, with the scope to show how this topic is considered as valid and has been approached inside the journal.

REPLY: We agree with the reviewer, a paper from METALS was added to the present work

Reviewer 2 Report

I would like to thank the author for this interesting manuscript titled “Effect of section thickness and solidification time on microstructural, mechanical and fatigue properties of solution strengthened ferritic ductile iron.”

There is a considerable number of papers published on the solution strengthened ferritic ductile iron discussing the microstructure, the mechanical and the fatigue properties. One of the recent article was published by the authors on Advanced Engineering Materials. It could be better to highlight more the novelty of this work to highlight the novelty.

Line 86 you introduced a numerical model used to identify the different regions with different solidification time. It would help to present in more details the numerical model to show how the regions where identified and provide details on the extension and location. More details on how the samples are cut out would help understand the analysis of the microstructure.

Line 98 is confusing, since the geometry described is rectangular cross section which is unusual for fatigue testing.

The thermal analysis technique is introduced without enough details. The set up and the description on how the observation were performed should be explained. Moreover, is it linked to the rest of the paper? If it is more details should be provided. The three figures 2, 3 and 4 are not fully explained.

The tensile testing should show the scatter in the experimental results. The discussion should be extended on the results and linked to the previous section. What difference has been observed on the tensile properties depending on the different conditions should be reported.

Reading the paper it is difficult to understand how many samples have been tested.

The solidification time and the positions shown in figure 5 should be presented with more details.

Line 249 how did you find the range for the initiating defects?

Figure 6 – the x axis label should be “Number of cycles”.

The fatigue curves at 50% should be better explained.

The defects reported have been already discussed and explained in the literature where the effects on the static and fatigue strength has also been discussed. The contribution to knowledge should be better explained.

Author Response

We really thank the Reviewer for the time spent in reading our work and comments aimed to improve it. All the suggestions were taken into account and the paper modified accordingly 

I would like to thank the author for this interesting manuscript titled “Effect of section thickness and solidification time on microstructural, mechanical and fatigue properties of solution strengthened ferritic ductile iron.”

REPLY: we thank a lot the Reviewer for the time spent in  reviewing the present work and his suggestions aimed to improve it.

There is a considerable number of papers published on the solution strengthened ferritic ductile iron discussing the microstructure, the mechanical and the fatigue properties. One of the recent article was published by the authors on Advanced Engineering Materials. It could be better to highlight more the novelty of this work to highlight the novelty.

Reply: we thank the Reviewer for the observation. The novelty  of the paper consists in heaving determined for the first time  and for this kind of alloy the correlation between the mechanical properties and the solidification times. This will overcome the limitations implicit in the mechanical properties-thickness relationship as specified in the updated version of the abstract, introduction and conclusions.

Line 86 you introduced a numerical model used to identify the different regions with different solidification time. It would help to present in more details the numerical model to show how the regions where identified and provide details on the extension and location. More details on how the samples are cut out would help understand the analysis of the microstructure.

The regions were identified according to the Standards mentioned in the paper. For simplicity, it was supposed that the thermal history measured by the virtual thermocouple at the centre of each region is sufficient to characterized the whole region. A better explanation was added to the text.

Line 98 is confusing, since the geometry described is rectangular cross section which is unusual for fatigue testing.

Reply: Rectangular cross section was more easy to obtain by machining in our laboratory; furthermore, rectangular cross section is allowed by Standards. The Used standard (ASTM E468-18)was added to the text 

The thermal analysis technique is introduced without enough details. The set up and the description on how the observation were performed should be explained. Moreover, is it linked to the rest of the paper? If it is, more details should be provided. The three figures 2, 3 and 4 are not fully explained.

Reply: Since the analysed alloy has never been characterized by thermal analysis, authors consider  the published thermal data useful to complete the characterization of the alloy itself (this was better specified in the text).   The used procedure is that used by foundries in all over the world, thus, its detailed description was considered redundant in the present work.  In any case more details were added in the text. 

The tensile testing should show the scatter in the experimental results. The discussion should be extended on the results and linked to the previous section. What difference has been observed on the tensile properties depending on the different conditions should be reported. 

Reply: scatters in results are added to table 2. The discussion was extended and more focused on the novelty of the paper

Reading the paper it is difficult to understand how many samples have been tested.

Reply: As specified in the paper, for tensile test, 5 samples for each condition were tested. For fatigue tests, a total of 60 and 30 specimens have been tested for type IV and type III sample, respectively

The solidification time and the positions shown in figure 5 should be presented with more details. 

Reply: The solidification time can be mapped as a result of the numerical simulation in NovaFlow & Solid. In the present work, the solidification time read in the centre of each specimens was used as representative of the specimen itself (this point was better specified in the manuscript). Figure 5 shows the position of each specimen’s cross section. By knowing the cross section dimensions and the dimension of the casting it should be easy to get details about the overall geometry.

Line 249 how did you find the range for the initiating defects? 

Reply: The range was obtained by analysing the fracture surface of different samples with particular reference to the weakest and the strongest ones

Figure 6 – the x axis label should be “Number of cycles”.

Reply: The x axis was modified according to the Reviewer’s suggestion

The fatigue curves at 50% should be better explained.

Reply:The survival probability p50% is thevalue of the probability that the sample will break at lower load values. A line of explanation was added to the text

The defects reported have been already discussed and explained in the literature where the effects on the static and fatigue strength has also been discussed. The contribution to knowledge should be better explained.

Reply: We thank the Reviewer for the suggestion. As  matter of fact, in the present work an effort was made to correlate the crack initiating defect dimension with the solidification time. As specified in the paper, we are checking if the ‘solidification time’ may be considered as an effective microstructure-influencing parameter (including defects). With a positive answer to this question, the static and fatigue properties of a generic casting can be predict, in each zone of it, simply by a thermal numerical computation, which is become a standard practice in modern foundries.This will allow to overcome drawbacks related to microstructural-mechanical numerical simulation. This important observation was added to the conclusions a we thank again the Reviewer for his contribution aimed to  improve manuscript.

Reviewer 3 Report

This is a good work on influence of thickness on structure of DCI.

The paper is well written.

Referee thinks that can be published as this

Author Response

We really thank the Reviewer for the time spent in reading our work and the positive comments about it

Round  2

Reviewer 1 Report

Everything is OK. Just very little remarks regarding the format:

- caption of fig. 8 moved in another page

- labels in figg. 8 and 9 show red underlines (probably useless)

Author Response

- caption of fig. 8 moved in another page

Reply: the layout was improved (Fig. 8 was moved in another page), thanks for the suggestion

- labels in figg. 8 and 9 show red underlines (probably useless)

Reply: the red lines are the markers that usually are indicated in a micrography to scale the dimensions of phases (they aren’t underlines) 

Reviewer 2 Report

Dear Authors, most of my comments have not been address in the manuscript and the answers provided are more to justify your work rather than improve the manuscript. I have suggested to provide more details about the probability of the fatigue curves (which is known to this reviewer) or about the microstructure and the mechanical propertes but I don't see many changes in the actual manuscript.

Author Response

Dear Reviewer,

where applicable we have insert the standard deviations of mechanical and microstructural data. About fatigue curve we have used a log normal distribution up to 2*10^6 cycles. Furthermore, as indicated by the Standard the scatter band was calculated with a confidence interval of 95% and the survival probability of 10 and 90%.

We are very sorry but we don't understand the reviewer comment. Please, specify better what do you mean with probability of the fatigue curve and microstructural and mechanical data. Thanks a lot.